# Comparison of alcohol consumption and tobacco use among Korean adolescents before and during the COVID-19 pandemic

**Wonseok Jeong**[ID]*

Department of Public Health, Graduate School, Seoul National University, Seoul, Republic of Korea

* wsjeong22@snu.ac.kr

**Data Availability Statement:** All KYRBWS files are available from the KYRBWS database (https://www.kdca.go.kr/yhs/).

**Funding:** This work was supported by the National Research Foundation of Korea (BK21 Center for

## Abstract

### Background

The COVID-19 pandemic has brought significant changes worldwide, and due to the strict "Social Distancing Plan" including school closures, Korean adolescents have experienced unprecedented changes in their lives. Considering the peer effect on adolescents' health behavior impacted due to the changes brought about by the pandemic, it would be interesting to explore differences in substance use in Korean adolescents. This study examines how these risk behaviors among Korean adolescents have changed before and during the COVID-19 pandemic.

### Methods

Korea Youth Risk Behavior Web-based Survey of 87,532 adolescents was used to collect the data for the period 2019, 2020, and 2021. The KYRBWS is conducted by a national institution which uses a stratified two-stage cluster sampling, and the data is statistically reliable and representative of the population. The Cochran-Armitage and Chi-squared test for linear and non-linear time trends, respectively, were calculated to assess the difference across the period (2019, 2020, 2021). Also, the odds ratios (ORs) with 95% CIs for current smoking status and current alcohol use status among 2020 and 2021 participants were compared with those of the 2019 participants using multiple logistic regression analysis.

### Results

The degree of current smoking status was lower in 2020 and 2021 participants than in the 2019 participants (2020: OR = 0.66, 95% CI = 0.61–0.71; 2021: OR = 0.66, 95% CI = 0.61–0.71). On the same token, current alcohol use status was also lower in the participants during the pandemic than those before the pandemic (2020: OR = 0.70, 95% CI = 0.66–0.73; 2021: OR = 0.70, 95% CI = 0.66–0.73).

### Conclusion

This study found that alcohol and tobacco use were reduced among Korean adolescents during the COVID-19 pandemic. Despite the decrease, future research on the potential effects of the COVID-19 pandemic on adolescents is warranted.

Integrative Response to Health Disasters, Graduate School of Public Health, Seoul National University) (NO.419 999 0514025). The funders had no role in study design, data collection and analysis, decision to publish, or preparation of the manuscript.

**Competing interests:** The authors have declared that no competing interests exist.

## Introduction

COVID-19 was first recognized in December 2019 in Wuhan, China, and due to its high contagiousness, more than 218,000 infected patients and 8,900 deaths had been reported by 18 March 2020, and the virus had reached 173 countries [1]. A total of 9786 confirmed cases of COVID-19 were reported in South Korea by 31 March 2020 [2]. Subsequently, on 11 March 2020, the World Health Organization announced that COVID-19 was a global pandemic, and to control the outbreak and minimize its spread, the South Korean government enforced a strict "Social Distancing Plan" which includes contact tracing, quarantine, social distancing, and school closures.

We know that adolescence is often associated with an increased need for social connection and peer acceptance, and a heightened sensitivity to peer influence [3]. The presence of friends increases the likelihood that adolescents will take certain risks; having friends who smoke or drink is one of the biggest predictors of adolescent engagement in these behaviors [4]. Furthermore, according to the previous study, peer differentiation strongly influences psychosocial maturity of adolescents, which can also eventually lead to problematic behaviors [5]. These factors refer that social distancing rules and the school closures due to the COVID-19 pandemic may have especially affected the risk and health behaviors of young people, more than those of the other age groups.

Furthermore, risk behaviors among adolescents are extremely maleficent since such behaviors often extend into adulthood, rendering them vulnerable in adulthood to preventable morbidities and mortalities [4]. For instance, adolescents who drink alcohol are more likely to have alcohol-related disorders, mental health problems, and chronic diseases in adulthood, while smoking in adolescents is positively associated with nicotine-dependent, cancer, and cardiovascular diseases in adulthood [6].

As such, the Korean government has enforced "stay-at-home" orders due to the COVID-19 outbreak, when the peers highly influence youths' risk behaviors. The behavior changes among adolescents are likely to happen and thus, youths' risk behaviors require special attention. Prior studies on adults or focused on different aspects of adolescents, have not explored the differential patterns of substance use among Korean adolescents due to the pandemic. This led us to compare the alcohol consumption and tobacco use before and during COVID-19 pandemic among South Korean adolescents.

## Materials and methods

### Data and study participants

The Korea Youth Risk Behavior Web-based Survey (KYRBWS) was used to collect information on the health-risk behaviors in Korean students (from seventh to the twelfth grades) for the period 2019, 2020, and 2021. The KYRBWS is conducted by a national institution, the Korea Centers for Disease Control and Prevention (KCDC), based on stratified two-stage random cluster sampling. 400 middle schools and 400 high schools are randomly selected within 131 districts in the first stage. In the second sampling stage, one class is selected from each grade within each chosen school [7]. Lastly, every student in the selected classes is surveyed except for school dropouts, those with special needs, and those who have difficulty in reading comprehension. Also, the written informed consent was obtained from each students' parents for the survey [8]. The survey data is statistically reliable and representative of the Korean adolescents. The KYRBWS was approved by the KCDC Institutional Review Board (2014-06EXP-02-P-A) in 2014. From 2015, the ethics approval for the KYRBWS was waived by the KCDC Institutional Review Board under the Bioethics & Safety Act and opened to the public. As the

data is publicly available, no approval related to the ethics of research was needed. All data is available from https://www.kdca.go.kr/yhs/home.jsp with the permission of the reasonable request.

Of the 167,099 individuals who participated in the surveys, those aged >19 years (n = 166,590) were excluded for the analysis. Nobody was excluded from the dependent variables, alcohol use status and smoking status, due to the missing data. Next, 87,532 participants remained due to the missing data from the variables 'educational level of father and mother.' There was no missing data in other covariates such as economic level, subjective health status, stress level, and BMI, leading the remaining 87,532 as the final study sample size.

## Dependent variables

Current smoking status and current alcohol use status were included as the main dependent variables in this study. Participants who smoked more than one conventional cigarette within a month was categorized as 'Yes' and those who have never smoked or have not smoked a single conventional cigarette within a month were categorized as 'No' in the variable, current smoking status. Similarly, participants who drank at least one glass of any type of alcohol within a month were categorized as 'Yes' while those who have never drunk or have not drunk a single glass of alcohol within a month were categorized as 'No' in the dependent variable, current alcohol use status. All answers were based on self-reported measures.

## Primary comparison variable

The primary independent variable was the COVID-19 pandemic. Since the first patient with COVID-19 in South Korea was diagnosed on February 19, 2020, the study compared alcohol consumption and tobacco use between 2019 (before), 2020 and 2021 (during) participants [9].

## Demographic and socioeconomic variables

The demographic characteristics included in the study were participants' age (14–16 (Middle School), 17–19 (High School)) and gender. Socioeconomic factors included participants' scholastic performance (low, middle, and high), subjective economic level (low, middle, and high) and education level of parents (middle school or less, high school, and college or over). Each participant's academic performance was self-reported and evaluated through one question: "In the past 12 months, how has your average academic performance been? [10]"

## Health-related variables

The health-related characteristics included the participants' self-reported health status (high, middle, and low), stress level (high, middle, and low) and body mass index (normal, and overweight & obese). Participants with > = 23 BMI were categorized as overweight & obese [11].

## Statistical analysis

Cochran-Armitage tests were used to test linear time-trend estimates, while Chi-squared tests were conducted to assess the association of demographic and socioeconomic characteristics of the study population with their substance use [12, 13]. Multiple logistic regression analyses were performed to compare the alcohol consumption and tobacco use before and after the COVID-19 pandemic after accounting for potential confounding variables, including demographic, socioeconomic, and health-related characteristics. Results are reported as odds ratios (OR) with a 95% confidence interval (CI). Differences were considered statistically significant

with a *p*-value of <0.05. All data analyses were conducted using SAS 9.4 software (version 9.4; SAS Institute Inc., Cary, NC, USA).

## Results

A total of 167,099 participants were included. After all exclusions, data from 87,532 participants were analyzed. Table 1 presents the general characteristics of the study population;

**Table 1. General characteristics of participants (n = 87532).**

| Variables | Total | | Participants, n (%) | | | | | |
|---|---|---|---|---|---|---|---|---|
| | | | 2019 | | 2020 | | 2021 | |
| | 87,532 | | 27,205 | | 30,366 | | 29,961 | |
| **Level of School** | | | | | | | | |
| Middle school | 46,359 | (53.0) | 13,979 | (51.4) | 15,900 | (52.4) | 16,480 | (55.0) |
| High school | 41,173 | (47.0) | 13,226 | (48.6) | 14,466 | (47.6) | 13,481 | (45.0) |
| **Sex** | | | | | | | | |
| Male | 40,404 | (46.2) | 12,409 | (45.6) | 14,165 | (46.6) | 13,830 | (46.2) |
| Female | 47,128 | (53.8) | 14,796 | (54.4) | 16,201 | (53.4) | 16,131 | (53.8) |
| **Scholastic performance** | | | | | | | | |
| High | 37,722 | (43.1) | 11,908 | (43.8) | 12,831 | (42.3) | 12,983 | (43.3) |
| Middle | 43,428 | (49.6) | 13,321 | (49.0) | 15,253 | (50.2) | 14,854 | (49.6) |
| Low | 6,382 | (7.3) | 1,976 | (7.3) | 2,282 | (7.5) | 2,124 | (7.1) |
| **Economic level of family** | | | | | | | | |
| High | 38,428 | (43.9) | 11,902 | (43.7) | 13,224 | (43.5) | 13,302 | (44.4) |
| Middle | 47,927 | (54.8) | 14,912 | (54.8) | 16,705 | (55.0) | 16,310 | (54.4) |
| Low | 1,177 | (1.3) | 391 | (1.4) | 437 | (1.4) | 349 | (1.2) |
| **Educational level of father** | | | | | | | | |
| College or over | 61,739 | (70.5) | 18,783 | (69.0) | 21,293 | (70.1) | 21,663 | (72.3) |
| High school | 24,323 | (27.8) | 7,929 | (29.1) | 8,551 | (28.2) | 7,843 | (26.2) |
| Middle school or less | 1,470 | (1.7) | 493 | (1.8) | 522 | (1.7) | 455 | (1.5) |
| **Educational level of mother** | | | | | | | | |
| College or over | 58,818 | (67.2) | 17,810 | (65.5) | 20,242 | (66.7) | 20,766 | (69.3) |
| High school | 27,589 | (31.5) | 9,000 | (33.1) | 9,744 | (32.1) | 8,845 | (29.5) |
| Middle school or less | 1,125 | (1.3) | 395 | (1.5) | 380 | (1.3) | 350 | (1.2) |
| **Subjective health status** | | | | | | | | |
| High | 60,468 | (69.1) | 19,233 | (70.7) | 21,537 | (70.9) | 19,698 | (65.7) |
| Middle | 20,044 | (22.9) | 5,903 | (21.7) | 6,586 | (21.7) | 7,555 | (25.2) |
| Low | 7,020 | (8.0) | 2,069 | (7.6) | 2,243 | (7.4) | 2,708 | (9.0) |
| **Stress level** | | | | | | | | |
| High | 17,227 | (19.7) | 5,107 | (18.8) | 6,536 | (21.5) | 5,584 | (18.6) |
| Middle | 37,574 | (42.9) | 11,234 | (41.3) | 13,626 | (44.9) | 12,714 | (42.4) |
| Low | 32,731 | (37.4) | 10,864 | (39.9) | 10,204 | (33.6) | 11,663 | (38.9) |
| **BMI** | | | | | | | | |
| Normal | 62,589 | (71.5) | 19,875 | (73.1) | 21,534 | (70.9) | 21,180 | (70.7) |
| Overweight & Obese | 24,943 | (28.5) | 7,330 | (26.9) | 8,832 | (29.1) | 8,781 | (29.3) |
| **Current smoking status** | | | | | | | | |
| Yes | 3,768 | (4.3) | 1,529 | (5.6) | 1,146 | (3.8) | 1,093 | (3.6) |
| No | 83,764 | (95.7) | 25,676 | (94.4) | 29,220 | (96.2) | 28,868 | (96.4) |
| **Current alcohol use status** | | | | | | | | |
| Yes | 9,988 | (11.4) | 3,876 | (14.2) | 3,109 | (10.2) | 3,003 | (10.0) |
| No | 77,544 | (88.6) | 23,329 | (85.8) | 27,257 | 27,257 | (89.8) | 3,003 (90.0) |

27,205 youths in 2019 were compared with 30,366 youths in 2020 and 29,961 youths in 2021. In 2019, 5.6% and 14.2% of the participants smoked conventional cigarettes and consumed alcohol, respectively. In 2020, the rates have decreased by 3.8% and 10.2%, and it continued to reduce until 3.6% and 10.0% in 2021. The 2019 participants had a lower severe stress level (39.9%) than that of 2020 (33.6%) and 2021 (38.9%) participants. Also, 2019 youths had the lowest overweight and obese rate of 14.7% compared to 15.9% and 16.7% from 2020 and 2021 participants.

Tables 2 and 3 present the trends in the proportion of Korean adolescents smoking cigarettes and taking alcohol respectively by their demographic, socioeconomic, and health-related variables. Considerable decreases in the levels of conducting both risk behaviors were observed

**Table 2. Trends of the current smokers (n = 3768).**

| | | | 2019 | | 2020 | | 2021 | | Total | | P-value for trend [a] |
|---|---|---|---|---|---|---|---|---|---|---|---|
| **Level of School** | | | | | | | | | | | |
| Middle school | | | 383 | (2.7) | 242 | (1.5) | 249 | (1.5) | 874 | (1.9) | < .0001[†] |
| High school | | | 1,146 | (8.7) | 904 | (6.3) | 844 | (6.3) | 2,894 | (7.0) | < .0001 |
| **Sex** | | | | | | | | | | | |
| Male | | | 1,037 | (8.4) | 783 | (5.5) | 709 | (5.1) | 2,529 | (6.3) | < .0001 |
| Female | | | 492 | (3.3) | 363 | (2.2) | 384 | (2.4) | 1,239 | (2.6) | < .0001[†] |
| **Scholastic performance** | | | | | | | | | | | |
| High | | | 409 | (3.4) | 239 | (1.9) | 232 | (1.8) | 880 | (2.3) | < .0001 |
| Middle | | | 806 | (6.1) | 614 | (4.0) | 606 | (4.1) | 2,026 | (4.7) | < .0001[†] |
| Low | | | 314 | (15.9) | 293 | (12.8) | 255 | (12.0) | 862 | (13.5) | 0.0003 |
| **Economic level of family** | | | | | | | | | | | |
| High | | | 614 | (5.2) | 493 | (3.7) | 436 | (3.3) | 1,543 | (4.0) | < .0001 |
| Middle | | | 867 | (5.8) | 611 | (3.7) | 621 | (3.8) | 2,099 | (4.4) | < .0001[†] |
| Low | | | 48 | (12.3) | 42 | (9.6) | 36 | (10.3) | 126 | (10.7) | 0.4465[†] |
| **Educational level of father** | | | | | | | | | | | |
| Middle school or less | | | 43 | (8.7) | 28 | (5.4) | 27 | (5.9) | 98 | (6.7) | 0.0756[†] |
| High school | | | 615 | (7.8) | 464 | (5.4) | 419 | (5.3) | 1,498 | (6.2) | < .0001 |
| College or over | | | 871 | (4.6) | 654 | (3.1) | 647 | (3.0) | 2,172 | (3.5) | < .0001 |
| **Educational level of mother** | | | | | | | | | | | |
| Middle school or less | | | 30 | (7.6) | 32 | (8.4) | 22 | (6.3) | 84 | (7.5) | 0.5442[†] |
| High school | | | 680 | (7.6) | 511 | (5.2) | 460 | (5.2) | 1,651 | (6.0) | < .0001[†] |
| College or over | | | 819 | (4.6) | 603 | (3.0) | 611 | (2.9) | 2,033 | (3.5) | < .0001 |
| **Subjective health status** | | | | | | | | | | | |
| High | | | 1,038 | (5.4) | 790 | (3.7) | 699 | (3.5) | 2,527 | (4.2) | < .0001 |
| Middle | | | 334 | (5.7) | 244 | (3.7) | 257 | (3.4) | 835 | (4.2) | < .0001 |
| Low | | | 157 | (7.6) | 112 | (5.0) | 137 | (5.1) | 406 | (5.8) | 0.0002[†] |
| **Stress level** | | | | | | | | | | | |
| Low | | | 229 | (4.5) | 213 | (3.3) | 149 | (2.7) | 591 | (3.4) | < .0001 |
| Middle | | | 531 | (4.7) | 415 | (3.0) | 383 | (3.0) | 1,329 | (3.5) | < .0001[†] |
| High | | | 769 | (7.1) | 518 | (5.1) | 561 | (4.8) | 1,848 | (5.6) | < .0001 |
| **BMI** | | | | | | | | | | | |
| Normal | | | 1,240 | (5.3) | 932 | (3.6) | 877 | (3.5) | 3,049 | (4.1) | < .0001 |
| Overweight & Obese | | | 289 | (7.2) | 214 | (4.4) | 216 | (4.3) | 719 | (5.2) | < .0001 |

[a] Based on Cochran-Armitage time trend analyses (for linear trend) and $\chi^2$ analyses
[†] (for nonlinear trend) with Rao-Scott adjustments to assess significant trends over time.

**Table 3. Trends of the current alcohol users (n = 9988).**

| | 2019 | | 2020 | | 2021 | | Total | | P-value for trend |
|---|---|---|---|---|---|---|---|---|---|
| **Level of School** | | | | | | | | | |
| Middle school | 1,036 | (7.4) | 867 | (5.5) | 936 | (5.7) | 2,839 | (6.1) | < .0001[†] |
| High school | 2,840 | (21.5) | 2,242 | (15.5) | 2,067 | (15.3) | 7,149 | (17.3) | < .0001 |
| **Sex** | | | | | | | | | |
| Male | 2,029 | (16.4) | 1,673 | (11.8) | 1,609 | (11.6) | 5,311 | (13.1) | < .0001 |
| Female | 1,847 | (12.5) | 1,436 | (8.9) | 1,394 | (8.6) | 4,677 | (9.9) | < .0001 |
| **Scholastic performance** | | | | | | | | | |
| High | 1,270 | (10.7) | 961 | (7.5) | 986 | (7.6) | 3,217 | (8.5) | < .0001[†] |
| Middle | 2,096 | (15.7) | 1,661 | (10.9) | 1,608 | (10.8) | 5,365 | (12.4) | < .0001 |
| Low | 510 | (25.8) | 487 | (21.3) | 409 | (19.3) | 1,406 | (22.0) | < .0001 |
| **Economic level of family** | | | | | | | | | |
| High | 1,582 | (13.3) | 1,245 | (9.4) | 1,198 | (9.0) | 4,025 | (10.5) | < .0001 |
| Middle | 2,198 | (14.7) | 1,787 | (10.7) | 1,729 | (10.6) | 5,714 | (11.9) | < .0001 |
| Low | 96 | (24.6) | 77 | (17.6) | 76 | (21.8) | 249 | (21.2) | 0.0483[†] |
| **Educational level of father** | | | | | | | | | |
| Middle school or less | 114 | (23.1) | 88 | (16.9) | 83 | (18.2) | 285 | (19.4) | 0.0314[†] |
| High school | 1,487 | (18.8) | 1,204 | (14.1) | 1,183 | (15.1) | 3,874 | (15.9) | < .0001 |
| College or over | 2,275 | (12.1) | 1,817 | (8.5) | 1,737 | (8.0) | 5,829 | (9.4) | < .0001 |
| **Educational level of mother** | | | | | | | | | |
| Middle school or less | 70 | (17.7) | 63 | (16.6) | 54 | (15.4) | 187 | (16.6) | 0.4012 |
| High school | 1,692 | (18.8) | 1,338 | (13.7) | 1,289 | (14.6) | 4,319 | (15.7) | < .0001[†] |
| College or over | 2,114 | (11.9) | 1,708 | (8.4) | 1,660 | (8.0) | 5,482 | (9.3) | < .0001 |
| **Subjective health status** | | | | | | | | | |
| High | 2,628 | (13.7) | 2,124 | (9.9) | 1,840 | (9.3) | 6,592 | (10.9) | < .0001 |
| Middle | 870 | (14.7) | 663 | (10.1) | 799 | (10.6) | 2,332 | (11.6) | < .0001[†] |
| Low | 378 | (18.3) | 322 | (14.4) | 364 | (13.4) | 1,064 | (15.2) | < .0001 |
| **Stress level** | | | | | | | | | |
| Low | 559 | (10.9) | 487 | (7.5) | 447 | (8.0) | 1,493 | (8.7) | < .0001[†] |
| Middle | 1,397 | (12.4) | 1,262 | (9.3) | 1,143 | (9.0) | 3,802 | (10.1) | < .0001 |
| High | 1,920 | (17.7) | 1,360 | (13.3) | 1,413 | (12.1) | 4,693 | (14.3) | < .0001 |
| **BMI** | | | | | | | | | |
| Normal | 3,147 | (13.7) | 2,502 | (9.8) | 2,356 | (9.4) | 8,005 | (10.9) | < .0001 |
| Overweight & Obese | 729 | (18.2) | 608 | (12.6) | 647 | (12.3) | 1,983 | (14.4) | < .0001 |

[a] Based on Cochran-Armitage time trend analyses (for linear trend) and $\chi^2$ analyses

[†] (for nonlinear trend) with Rao-Scott adjustments to assess significant trends over time.

during the pandemic among the majority of the adolescents categorized by their level of school, sex, scholastic performance, subjective health status, stress level, and BMI. Adolescents with low economic level of family and low educational level of parents did not present statistically significant differences in current smoking rates over the pandemic. Similarly, adolescents with low educational level of parents showed no indication of significant differences in the use of alcohol before and during the pandemic.

Table 4 shows the association between two risk behaviors and COVID-19 pandemic among Korean adolescents. The results are adjusted for age, sex, economic level, educational level of father and mother, scholastic performance, subjective health status, stress level and BMI. Participants in 2020 and 20201 were estimated with lower odd ratios of current smoking status

**Table 4. Logistic regression results on substance uses for 2019 vs 2020, 2019 vs 2021 participants.**

| Variables | 2020 | | | | 2021 | | | |
|---|---|---|---|---|---|---|---|---|
| | Adjusted OR [a] | 95% CI | | | Adjusted OR [a] | 95% CI | | |
| **Current smoking status** | | | | | | | | |
| No | 1.00 | | | | 1.00 | | | |
| Yes | 0.66* | (0.61 | – | 0.71) | 0.66* | (0.61 | – | 0.71) |
| **Current alcohol use status** | | | | | | | | |
| No | 1.00 | | | | 1.00 | | | |
| Yes | 0.70* | (0.66 | – | 0.73) | 0.70* | (0.66 | – | 0.73) |

Statistically significant was marked as

*. Abbreviations: CI, Confidence interval; OR, Odds ratio

[a] Adjusted for age, sex, economic level, educational level of father and mother, scholastic performance, subjective health status, stress level and BMI.

compared to the participants in 2019. These results were statistically significant (2020: OR = 0.66, 95% CI = 0.61–0.71; 2021: OR = 0.66, 95% CI = 0.61–0.71). Also, 2020 and 2021 participants presented with lower odd ratios of current alcohol use status compared to the participants in 2019. These results were statistically significant (2020: OR = 0.70, 95% CI = 0.66–0.73; 2021: OR = 0.70, 95% CI = 0.66–0.73).

Table 5 shows the results of a subgroup analysis between two risk behaviors and COVID-19 pandemic, focusing on sex and scholastic performance. Both male and female groups demonstrated lower odds of smoking cigarettes and taking alcohol during COVID-19. In addition, according to the scholastic performance, all groups (low, middle, and high) illustrated lower odds of smoking cigarettes and taking alcohol in the 2020 and 2021 participants than in the 2019 participants.

## Discussion

There is little empirical evidence on the changes of alcohol and tobacco consumption during the COVID-19 pandemic of Korean teenagers. This study describes the connections between COVID-19 pandemic and its subsequent "stay-at-home" orders and changes in substance uses of Korean adolescents, using demographic, socioeconomic, and health-related variables gained from the 2019, 2020, and 2021 KYRBWS data. There were positive associations with COVID-19 and a statistically significant decreased substance use; adolescents smoked less cigarettes and drank less alcohol compared to how they did before the pandemic. The lower odds of alcohol consumption and cigarettes use were consistent in all subgroups by both sex and scholastic performance.

Social distancing plan and its subsequent school closure due to the COVID-19 outbreak seem to be the reasons for the reduction in the substance uses of teenagers. Adolescence is associated with an increased need for social connection and therefore, presence of friends increases the likelihood of risk behaviors including smoking cigarettes and taking alcohol [3, 4]. Reduced peer pressure and importance of peer relationships from the school closure could have stopped youths from conducting unnecessary risk behaviors. According to the subgroup analysis results of risk behaviors and COVID-19 pandemic in Table 3, such peer pressure must have affected adolescents equally regardless of sex and educational performances. In addition, reduced stress due to the stay-at-home orders might also have reduced the substance uses of adolescents. According to the previous study, school closures can alleviate stress in youths by reducing academic burdens and school bullying, while there was a significant association between high perceived stress and cigarette use [14–16]. On the same token, stressful school

**Table 5. Logistic regression results on substance uses for 2019 vs 2020, 2019 vs 2021 participants by sex and scholastic performance.**

| Variables | 2020 | | | 2021 | | |
|---|---|---|---|---|---|---|
| | Adjusted OR [a] | 95% CI | | Adjusted OR [a] | 95% CI | |
| **Sex** | | | | | | |
| **Men (n = 40,404)** | | | | | | |
| Current smoking status | | | | | | |
| No | 1.00 | | | 1.00 | | |
| Yes | 0.63* | (0.57 | – | 0.70) | 0.61* | (0.55 – 0.68) |
| Current alcohol use status | | | | | | |
| No | 1.00 | | | 1.00 | | |
| Yes | 0.68* | (0.64 | – | 0.73) | 0.69* | (0.65 – 0.75) |
| **Women (n = 47,128)** | | | | | | |
| Current smoking status | | | | | | |
| No | 1.00 | | | 1.00 | | |
| Yes | 0.71* | (0.62 | – | 0.82) | 0.76* | (0.66 – 0.87) |
| Current alcohol use status | | | | | | |
| No | 1.00 | | | 1.00 | | |
| Yes | 0.71* | (0.66 | – | 0.76) | 0.70* | (0.66 – 0.76) |
| **Scholastic performance** | | | | | | |
| **High (n = 37,722)** | | | | | | |
| Current smoking status | | | | | | |
| No | 1.00 | | | 1.00 | | |
| Yes | 0.57* | (0.49 | – | 0.68) | 0.56* | (0.47 – 0.66) |
| Current alcohol use status | | | | | | |
| No | 1.00 | | | 1.00 | | |
| Yes | 0.72* | (0.65 | – | 0.78) | 0.74* | (0.68 – 0.81) |
| **Middle (n = 43,428)** | | | | | | |
| Current smoking status | | | | | | |
| No | 1.00 | | | 1.00 | | |
| Yes | 0.66* | (0.59 | – | 0.73) | 0.69* | (0.62 – 0.77) |
| Current alcohol use status | | | | | | |
| No | 1.00 | | | 1.00 | | |
| Yes | 0.66* | (0.62 | – | 0.71) | 0.67* | (0.63 – 0.72) |
| **Low (n = 6,382)** | | | | | | |
| Current smoking status | | | | | | |
| No | 1.00 | | | 1.00 | | |
| Yes | 0.79* | (0.66 | – | 0.94) | 0.73* | (0.61 – 0.88) |
| Current alcohol use status | | | | | | |
| No | 1.00 | | | 1.00 | | |
| Yes | 0.80* | (0.69 | – | 0.92) | 0.70* | (0.60 – 0.81) |

Statistically significant was marked as

*. Abbreviations: CI, Confidence interval; OR, Odds ratio

[a] Adjusted for age, sex, economic level, educational level of father and mother, scholastic performance, subjective health status, stress level and BMI.

environment due to South Korea's educational fever can also place students at risk of alcohol [17]. Lastly, increased family warmth and connectedness due to the social distancing plan might have served as a protective factor against many of the risky behaviors engaged in by adolescents [18]. For example, the prior study presents that in comparison with children who

spent more time with their parents, those who spent less time were more inclined to start smoking and try alcohol [19]. Considering that 21.4% of school-aged children found the parent-children discussions to be more satisfying during the school closures, the argument is pretty valid [20].

Yet, the current study's limitations should be noted. First, the data in this study are based on self-reported measures, and health status measurements might be subject to recall bias. Therefore, caution should be taken when interpreting these results. Also, due to this study's cross-sectional design, cause, and effect, as well as the direction of the relationships observed, could not be determined. Third, cultural aspects could have influenced the impact of the COVID-19 pandemic on smoking and alcohol use status of Korean adolescents. Lastly, adolescents were considered as current smokers and drinkers if they have smoked or drank at least once within the past thirty days. The exact number of cigarettes smoked, and amount of alcohol consumed were not involved in the investigation.

Despite these limitations, our study does possess several strengths. The KYRBWS is conducted by a national institution based on random cluster sampling, and therefore, the data gained from it is statistically reliable and representative in comparison to surveys performed by private institutions. Moreover, as this study was conducted for over three years, compared to most recent COVID-19 research which only include one or two years, the representativeness of the sample was improved upon. Lastly, many covariates, including age, sex, economic level, educational levels of parents, scholastic performance, subjective health status, and BMI, were included to reduce the possible confounding effects.

## Conclusions

Along with the severe health consequences from the pandemic, long-term health consequences from the risk behaviors, especially among adolescents, are significant public health concern. This study found that the substance uses (alcohol and tobacco usage) were reduced among Korean adolescents during the COVID-19 pandemic period regardless of sex and scholastic performance. Despite the decrease, however, we must not belittle the prevalence of risk behaviors among adolescents and the potential effects of the COVID-19 pandemic on adolescents require additional follow-up studies.

## Supporting information

**S1 Table.**
(DOCX)

## Author Contributions

**Conceptualization:** Wonseok Jeong.

**Data curation:** Wonseok Jeong.

**Methodology:** Wonseok Jeong.

**Writing – original draft:** Wonseok Jeong.

**Writing – review & editing:** Wonseok Jeong.

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
