## [Decision Letter · Decision Letter 0]

10 Oct 2022

PONE-D-22-20225Comparison of Alcohol Consumption and Tobacco Use Among Korean Adolescents Before and During the COVID-19 PandemicPLOS ONE

Dear Dr. Jeong,

Thank you for submitting your manuscript to PLOS ONE. After careful consideration, we feel that it has merit but does not fully meet PLOS ONE’s publication criteria as it currently stands. Therefore, we invite you to submit a revised version of the manuscript that addresses the points raised during the review process.

We look forward to receiving your revised manuscript.

Kind regards,

Chandan Kumar, Ph.D.

Academic Editor

PLOS ONE

Journal Requirements:

Additional Editor Comments:

The manuscript has been reviewed by two independent reviewers, and their comments are given below. You are requested to screen all the comments/suggestions forwarded by the reviewers and incorporate them accordingly to further improve your manuscript. While submitting the revised manuscript, also submit the point-to-point response to the reviewers’ comments/suggestions.

Comments from Reviewer 1: Minor Revision

This study compared the changes in risk behaviors in Korean adolescents, such as smoking and drinking, pre- and during the COVID-19 pandemic. The study is well thought-out and the manuscript is clear and concise, however I do have some minor comments that can be taken into account before the paper is further considered for publication.

1. Please revise the results section in the abstract once again. There are terms such as ‘ex-smoker’, ‘single smoker’ and ‘dual smoker’ used in the abstract which cannot be found in the main manuscript.

2. It would be good if there was more information regarding the smoking and drinking variables, such as the exact number of cigarettes smoked or drinks consumed. The author also mentioned in the discussion that adolescents smoked fewer cigarettes and drank less alcohol, which would be accurate if variables quantifying the amount of tobacco and alcohol used were included in the analysis, if available. Otherwise, I would suggest including it in the limitations of the study.

3. Table 4 shows the subgroup analysis results of risk behaviors and COVID-19 pandemic, stratified by sex and scholastic performance, yet there is no mention of the significance of these results in the discussion section. Please write a line or two regarding these points in the discussion.

4. Since the author mentioned stress as a possible factor behind heightened alcohol and tobacco use in the discussion, is it possible to include ‘stress level’ in the subgroup analysis as well and check whether it is statistically meaningful?

5. “Reduced peer pressure and importance of peer relationships from the school closure could have stopped youths from conducting unnecessarily risk behaviors.” Please change ‘unnecessarily’ to ‘unnecessary’.

Comments from Reviewer 2: Major Revision

Overall Comments

The COVID-19 pandemic has initiated new challenges related to health-risk behaviors. This is an important study that compares the current smoking/drinking behavior of adolescents before and after the outbreak of the pandemic. The data source used in the study is also interesting, and likely to provide insight into the health effect of COVID-19. Overall, the manuscript is interesting and suitable for the journal's aims and scopes.

Title

The title did not adequately represent the methods and findings. In abstract, the outcome measures are current smoking and current drinking status. Whereas, the outcome measures written in the result are presented as “Non-Drinker” or “Non-Smoker.” Please update the title accordingly.

Introduction

Please clarify the background section further.

What is the definition of adolescence? To justify the need for this study, please discuss the plausibility – how is the stage of adolescence associated with risk behaviors? Why are they vulnerable to peer-pressure? Evidence regarding psychosocial development in adolescence is required.

The rationale on which the study was based is not strong enough - Why is it necessary to understand the role of social relationships in risk-behaviors among adolescents?

Data and study population

The total KYRBS survey participants comprised 167,099 school students, but approximately half of the survey participants were excluded from the study sample. Please provide the point-by-point exclusion criteria – In what process, did the survey participants drop out the most and why?

Variable

Please clarify the dependent variable of the study. How is the current smoking and drinking status measured? Was it examined? or based on questionnaire?

The offical outbreak of COVID-19 assigned by WHO is Feb 2020. Is the period of KYRBS data collection appropriate for the study?

Table 1

In the statistical analysis section, it is stated that the chi-square test was conducted to analyze the general characteristics of the study population. However, Table 1 does not present the result of this test. Please provide the result of chi-square test for independence. Is the distribution of categorized group differ from one another with the significant P-value?

Table 2

Please specify the performed test in the title of Tables 2 & 3.

The title of the study and the result do not align. I recommend estimating the OR for “Current Drinking Status” and “Current Smoking Status.” With the year 2019 served as reference, please provide OR for 2020 and 2021 – How much “less” likely the drinking or smoking is likely to occur relative to 2019? Please update the table and text accordingly.

Discussion

The statement “To our knowledge, this is the first study to present the changes of alcohol and tobacco consumption during the COVID-19 pandemic of Korean teenagers” is not true. Please remove the statement or rephrase it to “there is little empirical evidence on …”

Please cite for more information. The discussion part can be better written with proper explanations and justifications of the data.

Reviewers' comments:

Reviewer's Responses to Questions

**Comments to the Author**

1. Is the manuscript technically sound, and do the data support the conclusions?

Reviewer #1: Yes

Reviewer #2: Yes

2. Has the statistical analysis been performed appropriately and rigorously? 

Reviewer #1: I Don't Know

Reviewer #2: Yes

3. Have the authors made all data underlying the findings in their manuscript fully available?

Reviewer #1: Yes

Reviewer #2: Yes

4. Is the manuscript presented in an intelligible fashion and written in standard English?

Reviewer #1: Yes

Reviewer #2: Yes

5. Review Comments to the Author

Reviewer #1: Review

This study compared the changes in risk behaviors in Korean adolescents, such as smoking and drinking, pre- and during the COVID-19 pandemic. The study is well thought-out and the manuscript is clear and concise, however I do have some minor comments that can be taken into account before the paper is further considered for publication.

1. Please revise the results section in the abstract once again. There are terms such as ‘ex-smoker’, ‘single smoker’ and ‘dual smoker’ used in the abstract which cannot be found in the main manuscript.

2. It would be good if there was more information regarding the smoking and drinking variables, such as the exact number of cigarettes smoked or drinks consumed. The author also mentioned in the discussion that adolescents smoked less cigarettes and drank less alcohol, which would be accurate if variables quantifying the amount of tobacco and alcohol used were included in the analysis, if available. Otherwise, I would suggest to include it in the limitations of the study.

3. Table 4 shows the subgroup analysis results of risk behaviors and COVID-19 pandemic, stratified by sex and scholastic performance, yet there is no mention of the significance of these results in the discussion section. Please write a line or two regarding these points in the discussion.

4. Since the author mentioned stress as a possible factor behind heightened alcohol and tobacco use in the discussion, is it possible to include ‘stress level’ in the subgroup analysis as well and check whether it is statistically meaningful?

5. “Reduced peer pressure and importance of peer relationships from the school closure could have stopped youths from conducting unnecessarily risk behaviors.” Please change ‘unnecessarily’ to ‘unnecessary’.

Reviewer #2: Overall Comments

The COVID-19 pandemic has initiated new challenges related to health-risk behaviors. This is an important study that compares a current smoking/drinking behavior of adolescents before and after the outbreak of the pandemic. The data source that used in the study is also interesting, and likely to provide insight on the health effect of COVID-19. Overall, the manuscript is interesting and suitable for the journal's aims and scopes.

Title

The title did not adequately represent the methods and findings. In abstract, the outcome measures are current smoking and current drinking status. Whereas, the outcome measures written in the result are presented as “Non-Drinker” or “Non-Smoker.” Please update the title accordingly.

Introduction

Please clarify the background section further.

What is the definition of adolescence? To justify the need of this study, please discuss the plausibility – how is the stage of adolescence associated with risk behaviors? Why are they vulnerable to peer-pressure? Evidence regarding psychosocial development in adolescence is required.

The rationale on which the study was based is not strong enough - Why it is necessary to understand the role of social relationship in risk-behaviors among adolescents?

Data and study population

The total KYRBS survey participants comprised of 167,099 school students, but approximately half of the survey participants were exlcuded from the study sample. Please provide the point-by-point exclusion criteria – In what process, did the survey participants drop out the most and why?

Variable

Please clarify the dependent variable of the study. How is the current smoking and drinking status measured? Was it examined? or based on questionnaire?

The offical outbreak of COVID-19 assigned by WHO is Feb 2020. Is the period of KYRBS data collection appropriate for the study?

Table 1

In the statistical analysis section, it is stated that the chi-square test was conducted to analyze the general characteristics of the study population. However, Table 1 does not present the result of this test. Please provide the result of chi-square test for independence. Is the distribution of categorized group differ from one another with the significant P-value?

Table 2

Please specify the performed test in the title of Table 2 & 3.

The title of the study and the result do not align. I recommend estimating the OR for “Current Drinking Status” and “Current Smoking Status.” With the year 2019 served as reference, please provide OR for 2020 and 2021 – How much “less” likely the drinking or smoking is likely to occur relative to 2019? Please update the table and text accordingly.

Discussion

The statement “To our knowledge, this is the first study to present the changes of alcohol and tobacco consumption during the COVID-19 pandemic of Korean teenagers” is not true. Please remove the statement or rephrase it to “there is little empirical evidence on …”

Please cite for more information. Discussion part can be better written with proper explanations and justifications of the data.

6. PLOS authors have the option to publish the peer review history of their article (what does this mean?). If published, this will include your full peer review and any attached files.

Reviewer #1: No

Reviewer #2: No

---

## [Author Response · Author response to Decision Letter 0]

20 Oct 2022

I was pleased to have the opportunity to revise my paper. In revising the paper, I have carefully considered your comments and suggestions. As instructed, I have attempted to explain the changes made in reaction to all of the reviewers’ comments. The reviewers’ comments were very helpful overall, and I appreciate the constructive feedback on my original submission. After addressing the issues raised, I feel the quality of the paper has greatly improved and I hope you agree. My response to each comment is as follows, and I attach a revision note with the highlighted, revised sections of the manuscript. Again, thank you for the valuable and helpful comments.

Response to Reviewer #1’s comments

The study is well thought-out and the manuscript is clear and concise, however I do have some minor comments that can be taken into account before the paper is further considered for publication.

Thank you for your great efforts in reviewing my manuscript. After the review, I totally revised the manuscript. 

Minor Revision: Please revise the results section in the abstract once again. There are terms such as ‘ex-smoker’, ‘single smoker’ and ‘dual smoker’ used in the abstract which cannot be found in the main manuscript.

Response: Thank you for your comment. We totally agree with your comment. We changed the conclusion part of abstract by changing the sentence from “This study found that ex-smoker and dual smoker had inverse associations with the COVID-19 pandemic among Korean adolescents.” into “This study found that alcohol consumption and tobacco use had inverse associations with the COVID-19 pandemic among Korean adolescents.” (revised manuscript, line 34~35) Thank you so much for your careful revision which allowed us to find our mistakes. 

Revised manuscript, line 34~35: This study found that alcohol consumption and tobacco use had inverse associations with the COVID-19 pandemic among Korean adolescents.

Minor comments: It would be good if there was more information regarding the smoking and drinking variables, such as the exact number of cigarettes smoked or drinks consumed. The author also mentioned in the discussion that adolescents smoked fewer cigarettes and drank less alcohol, which would be accurate if variables quantifying the amount of tobacco and alcohol used were included in the analysis, if available. Otherwise, I would suggest including it in the limitations of the study.

Response: Thank you for your comment. We totally agree with your comment. We did not consider exact frequency of smoking/drinking since the studying populations. We, however, defined current smokers/drinkers as those who drank/smoked at least once withing a past month which would be still very detrimental to adolescents. Also, following your kind advise, we added the following sentence in the limitations of the study. 

Revised manuscript, line 202~204: Lastly, adolescents were considered as current smokers and drinkers if they have smoked or drank at least once within the past thirty days. The exact number of cigarettes smoked and drinks consumed were not involved in the investigation. 

Minor comments: Table 4 shows the subgroup analysis results of risk behaviors and COVID-19 pandemic, stratified by sex and scholastic performance, yet there is no mention of the significance of these results in the discussion section. Please write a line or two regarding these points in the discussion.

Response: Thank you for your comment. I totally agree with your comment. I added a sentence, “According to the subgroup analysis results of risk behaviors and COVID-19 pandemic in table 4, such peer pressure must have affected adolescents equally regardless of sex and educational performances.” (revised manuscript, line 184~186) Thank you for your meaningful comment.

Revised manuscript, line 184~186: According to the subgroup analysis results of risk behaviors and COVID-19 pandemic in table 4, such peer pressure must have affected adolescents equally regardless of sex and educational performances.

Minor comments: Since the author mentioned stress as a possible factor behind heightened alcohol and tobacco use in the discussion, is it possible to include ‘stress level’ in the subgroup analysis as well and check whether it is statistically meaningful?

Response: Thank you for your comment. We totally agree with your comment. have conducted a subgroup analysis including ‘stress level,’ but the result did not necessarily assist my comment on the reasonings for alcohol and tobacco use; no matter the stress levels every adolescent showed lower risk of smoking/drinking. Still, we appreciate your meaningful comment and providing me a chance to more deeply investigate our study.

Minor comments: “Reduced peer pressure and importance of peer relationships from the school closure could have stopped youths from conducting unnecessarily risk behaviors.” Please change ‘unnecessarily’ to ‘unnecessary’.

Response: Thank you for your comment. Thank you for correcting my grammatical error. The sentence has been changed as the below. 

Revised manuscript, line 182~184: Reduced peer pressure and importance of peer relationships from the school closure could have stopped youths from conducting unnecessary risk behaviors.

Response to Reviewer #2’s comments

This is an important study that compares the current smoking/drinking behavior of adolescents before and after the outbreak of the pandemic. The data source used in the study is also interesting, and likely to provide insight into the health effect of COVID-19. Overall, the manuscript is interesting and suitable for the journal's aims and scopes.

Thank you for your great efforts in reviewing our manuscript. I appreciate your comments and fully understand your concerns for the problems you have mentioned.

Title:

Major comments: The title did not adequately represent the methods and findings. In abstract, the outcome measures are current smoking and current drinking status. Whereas, the outcome measures written in the result are presented as “Non-Drinker” or “Non-Smoker.” Please update the title accordingly.

Response: Thank you for your comment. I totally understand what you mean. I changed the outcome measures written in the result as “Current Smoker” and “Current Drinker” in order to match the current title. 

Introduction:

Major comments: Please clarify the background section further.

Response: Thank you for your comment. We totally agree with your comment. We clarified background section by adding a following sentence: “As of 31 March 2020, a total of 9786 confirmed cases with COVID-19 have been reported in South Korea as well..” smoking on dyslipidemia in South Korean adults.” (revised manuscript, line 40~42) 

Major comments: What is the definition of adolescence? To justify the need for this study, please discuss the plausibility – how is the stage of adolescence associated with risk behaviors? Why are they vulnerable to peer-pressure? Evidence regarding psychosocial development in adolescence is required.

Response: Thank you for your comment. I totally agree with your comment. I added a sentence which can justify how the stage of adolescence is associated with risk behaviors as follows: “Furthermore, according to the pervious study, peer differentiation strongly influences psychosocial maturity of adolescents, which can also eventually lead to problematic behaviors.” Furthermore, regarding the definition of adolescence, further description has been added in the method section. Thank you again for your meaningful advise. 

Revised manuscript, line 50~52: Furthermore, according to the pervious study, peer differentiation strongly influences psychosocial maturity of adolescents, which can also eventually lead to problematic behaviors.

Major comments: The rationale on which the study was based is not strong enough - Why is it necessary to understand the role of social relationships in risk-behaviors among adolescents?

Response: Thank you for your comment. By adding a sentence from your previous advise, I believe this problem also might have been solved. 

Revised manuscript, line 50~52: Furthermore, according to the pervious study, peer differentiation strongly influences psychosocial maturity of adolescents, which can also eventually lead to problematic behaviors.

Data and study population:

Major comments: The total KYRBS survey participants comprised 167,099 school students, but approximately half of the survey participants were excluded from the study sample. Please provide the point-by-point exclusion criteria – In what process, did the survey participants drop out the most and why?

Response: Thank you for your comment. I totally agree with your comment. I carefully examined where the exclusion has been made, and added a sentence to the manuscript explaining it. The added sentence is “ next, among 166,590, 87,532 remained due to the missing data from the variable ‘educational level of father and mother.’ Finally, there was no missing data in other covariates such as economic level, subjective health status, stress level, and BMI, leading the remaining 87,532 as the final study sample size.”

Revised manuscript, line 88~91: Next, among 166,590, 87,532 remained due to the missing data from the variable ‘educational level of father and mother.’ Finally, there was no missing data in other covariates such as economic level, subjective health status, stress level, and BMI, leading the remaining 87,532 as the final study sample size.

Variables:

Major comments: Please clarify the dependent variable of the study. How is the current smoking and drinking status measured? Was it examined? or based on questionnaire?

Response: Thank you for your comment. I have written in the limitation that the smoking and drinking status was based on the survey. To enhance an understanding of the study, I also have added a following sentence in method section

Revised manuscript, line 112~113: All answers were based on self-reported measures.

Major comments: The official outbreak of COVID-19 assigned by WHO is Feb 2020. Is the period of KYRBS data collection appropriate for the study?

Response: Thank you for your comment. Even though the official outbreak of COVID-19 pandemic assigned by WHO is Feb 2020, the first recognition of the pandemic was in December 2019, which means that the COVID-19 has impacted Koreans from the very beginning of 2020. In addition, I could find various other researches which considered 2020 as a start year of COVID-19. Still, I appreciate your careful advise. 

Table 1:

Major comments: In the statistical analysis section, it is stated that the chi-square test was conducted to analyze the general characteristics of the study population. However, Table 1 does not present the result of this test. Please provide the result of chi-square test for independence. Is the distribution of categorized group differ from one another with the significant P-value?

Response: Thank you for your comment. Sorry for making a confusion and thank you for pointing out our mistake. We added a section which tells the p-value of result of chi-square test in the right side of table 1. Again, I appreciate your advise. 

Table 2:

Major comments: Please specify the performed test in the title of Tables 2 & 3.

Response: Thank you for your comment. I totally agree with your comment. Following your advise, I changed the title of Tables 2,3, and 4 to “Table 2. Logistic Regression Results on Drinking for 2019 vs 2020, 2019 vs 2021 Participants,” “Table 3. Logistic Regression Results on Smoking for 2019 vs 2020, 2019 vs 2021 Participants,” and “Table 4. Logistic Regression Results on Substance Uses for 2019 vs 2020, 2019 vs 2021 Participants By Sex and Scholastic Performance,” accordingly. 

Major comments: The title of the study and the result do not align. I recommend estimating the OR for “Current Drinking Status” and “Current Smoking Status.” With the year 2019 served as reference, please provide OR for 2020 and 2021 – How much “less” likely the drinking or smoking is likely to occur relative to 2019? Please update the table and text accordingly.

Response: Thank you for your comment. I totally agree with your comment. Following your advise, we have changed the result to show how much less likely the drinking or smoking would occur. Also, other parts in the manuscript were changed accordingly. 

Discussion:

Major comments: The statement “To our knowledge, this is the first study to present the changes of alcohol and tobacco consumption during the COVID-19 pandemic of Korean teenagers” is not true. Please remove the statement or rephrase it to “there is little empirical evidence on …”

Response: Thank you for your comment. I totally agree with your comment. I changed the sentence into “There is little empirical evidence on the changes of alcohol and tobacco consumption during the COVID-19 pandemic of Korean teenagers.”

 Revised manuscript, line 174~175: There is little empirical evidence on the changes of alcohol and tobacco consumption during the COVID-19 pandemic of Korean teenagers.

Major comments: Please cite for more information. The discussion part can be better written with proper explanations and justifications of the data.

Response: Thank you for your comment. Since all the necessary information was cited in the discussion section, I have added an explanation and a citation on the introduction instead. Citation is very important and thank you very much for reminding me of it.

---

## [Editor Report · Decision Letter 1]

4 Nov 2022

PONE-D-22-20225R1Comparison of Alcohol Consumption and Tobacco Use Among Korean Adolescents Before and During the COVID-19 PandemicPLOS ONE

Dear Dr. Jeong,

Thank you for submitting your manuscript to PLOS ONE. After careful consideration, the revised manuscript still needs to consider a few aspects to further improve the manuscript to fully meet PLOS ONE’s publication criteria as it currently stands. Therefore, we invite you to submit a revised version of the manuscript that addresses the points raised during the review process.

Academic Editor's Comments:

1. Please consider using a standard terminology, such as, "alcohol use" than "drinking" through out the manuscript.

2. I also found that the manuscript needs to undergo a through language editing for better clarity of the sentences.

3. To assess the difference across the time period (2019 vs 2020 vs 2021), the p-value for the time trend should be calculated. The Cochran-Armitage and Chi-squared test for linear and non-linear time trends, respectively, can be calculated. 

Please refer to the following reference:

https://journals.plos.org/plosone/article?id=10.1371/journal.pone.0069094

4. In Table 4, please change the ORs as per the results discussed. Changing the reference category from "Yes" to "No" would suffice. That would provide lower ORs in 2020/2021 compared to 2019.

5. Other minor editing required has been highlighted in the attached manuscripts use track change mode and adding comments. Please submit your revised manuscript by Dec 19 2022 11:59PM. If you will need more time than this to complete your revisions, please reply to this message or contact the journal office at plosone@plos.org. Please include the following items when submitting your revised manuscript:A rebuttal letter that responds to each point raised by the academic editor and reviewer(s). You should upload this letter as a separate file labeled 'Response to Reviewers'.A marked-up copy of your manuscript that highlights changes made to the original version. You should upload this as a separate file labeled 'Revised Manuscript with Track Changes'.An unmarked version of your revised paper without tracked changes. You should upload this as a separate file labeled 'Manuscript'.If applicable, we recommend that you deposit your laboratory protocols in protocols.io to enhance the reproducibility of your results. Protocols.io assigns your protocol its own identifier (DOI) so that it can be cited independently in the future. For instructions see: https://journals.plos.org/plosone/s/submission-guidelines#loc-laboratory-protocols. Additionally, PLOS ONE offers an option for publishing peer-reviewed Lab Protocol articles, which describe protocols hosted on protocols.io. Read more information on sharing protocols at https://plos.org/protocols?utm_medium=editorial-email&utm_source=authorletters&utm_campaign=protocols.

We look forward to receiving your revised manuscript.

Kind regards,

Chandan Kumar, Ph.D.

Academic Editor

PLOS ONE
---

## [Author Response · Author response to Decision Letter 1]

5 Nov 2022

I was pleased to have the opportunity to revise my paper. In revising the paper, I have carefully considered your comments and suggestions. As instructed, I have attempted to explain the changes made in reaction to all of the reviewers and academic editor’s comments. The comments were very helpful overall, and I appreciate the constructive feedback on my original submission. After addressing the issues raised, I feel the quality of the paper has greatly improved and I hope you agree. My response to each comment is as follows, and I attach a revision note with the highlighted, revised sections of the manuscript. Again, thank you for the valuable and helpful comments.

Response to Reviewer #1’s comments

The study is well thought-out and the manuscript is clear and concise, however I do have some minor comments that can be taken into account before the paper is further considered for publication.

Thank you for your great efforts in reviewing my manuscript. After the review, I totally revised the manuscript. 

Minor Revision: Please revise the results section in the abstract once again. There are terms such as ‘ex-smoker’, ‘single smoker’ and ‘dual smoker’ used in the abstract which cannot be found in the main manuscript.

Response: Thank you for your comment. We totally agree with your comment. We changed the conclusion part of abstract by changing the sentence from “This study found that ex-smoker and dual smoker had inverse associations with the COVID-19 pandemic among Korean adolescents.” into “This study found that alcohol consumption and tobacco use had inverse associations with the COVID-19 pandemic among Korean adolescents.” (revised manuscript, line 34~35) Thank you so much for your careful revision which allowed us to find our mistakes. 

Revised manuscript, line 34~35: This study found that alcohol consumption and tobacco use had inverse associations with the COVID-19 pandemic among Korean adolescents.

Minor comments: It would be good if there was more information regarding the smoking and drinking variables, such as the exact number of cigarettes smoked or drinks consumed. The author also mentioned in the discussion that adolescents smoked fewer cigarettes and drank less alcohol, which would be accurate if variables quantifying the amount of tobacco and alcohol used were included in the analysis, if available. Otherwise, I would suggest including it in the limitations of the study.

Response: Thank you for your comment. We totally agree with your comment. We did not consider exact frequency of smoking/drinking since the studying populations. We, however, defined current smokers/drinkers as those who drank/smoked at least once withing a past month which would be still very detrimental to adolescents. Also, following your kind advise, we added the following sentence in the limitations of the study. 

Revised manuscript, line 202~204: Lastly, adolescents were considered as current smokers and drinkers if they have smoked or drank at least once within the past thirty days. The exact number of cigarettes smoked and drinks consumed were not involved in the investigation. 

Minor comments: Table 4 shows the subgroup analysis results of risk behaviors and COVID-19 pandemic, stratified by sex and scholastic performance, yet there is no mention of the significance of these results in the discussion section. Please write a line or two regarding these points in the discussion.

Response: Thank you for your comment. I totally agree with your comment. I added a sentence, “According to the subgroup analysis results of risk behaviors and COVID-19 pandemic in table 4, such peer pressure must have affected adolescents equally regardless of sex and educational performances.” (revised manuscript, line 184~186) Thank you for your meaningful comment.

Revised manuscript, line 184~186: According to the subgroup analysis results of risk behaviors and COVID-19 pandemic in table 4, such peer pressure must have affected adolescents equally regardless of sex and educational performances.

Minor comments: Since the author mentioned stress as a possible factor behind heightened alcohol and tobacco use in the discussion, is it possible to include ‘stress level’ in the subgroup analysis as well and check whether it is statistically meaningful?

Response: Thank you for your comment. We totally agree with your comment. have conducted a subgroup analysis including ‘stress level,’ but the result did not necessarily assist my comment on the reasonings for alcohol and tobacco use; no matter the stress levels every adolescent showed lower risk of smoking/drinking. Still, we appreciate your meaningful comment and providing me a chance to more deeply investigate our study.

Minor comments: “Reduced peer pressure and importance of peer relationships from the school closure could have stopped youths from conducting unnecessarily risk behaviors.” Please change ‘unnecessarily’ to ‘unnecessary’.

Response: Thank you for your comment. Thank you for correcting my grammatical error. The sentence has been changed as the below. 

Revised manuscript, line 182~184: Reduced peer pressure and importance of peer relationships from the school closure could have stopped youths from conducting unnecessary risk behaviors.

Response to Reviewer #2’s comments

This is an important study that compares the current smoking/drinking behavior of adolescents before and after the outbreak of the pandemic. The data source used in the study is also interesting, and likely to provide insight into the health effect of COVID-19. Overall, the manuscript is interesting and suitable for the journal's aims and scopes.

Thank you for your great efforts in reviewing our manuscript. I appreciate your comments and fully understand your concerns for the problems you have mentioned.

Title:

Major comments: The title did not adequately represent the methods and findings. In abstract, the outcome measures are current smoking and current drinking status. Whereas, the outcome measures written in the result are presented as “Non-Drinker” or “Non-Smoker.” Please update the title accordingly.

Response: Thank you for your comment. I totally understand what you mean. I changed the outcome measures written in the result as “Current Smoker” and “Current Drinker” in order to match the current title. 

Introduction:

Major comments: Please clarify the background section further.

Response: Thank you for your comment. We totally agree with your comment. We clarified background section by adding a following sentence: “As of 31 March 2020, a total of 9786 confirmed cases with COVID-19 have been reported in South Korea as well..” smoking on dyslipidemia in South Korean adults.” (revised manuscript, line 40~42) 

Major comments: What is the definition of adolescence? To justify the need for this study, please discuss the plausibility – how is the stage of adolescence associated with risk behaviors? Why are they vulnerable to peer-pressure? Evidence regarding psychosocial development in adolescence is required.

Response: Thank you for your comment. I totally agree with your comment. I added a sentence which can justify how the stage of adolescence is associated with risk behaviors as follows: “Furthermore, according to the pervious study, peer differentiation strongly influences psychosocial maturity of adolescents, which can also eventually lead to problematic behaviors.” Furthermore, regarding the definition of adolescence, further description has been added in the method section. Thank you again for your meaningful advise. 

Revised manuscript, line 50~52: Furthermore, according to the pervious study, peer differentiation strongly influences psychosocial maturity of adolescents, which can also eventually lead to problematic behaviors.

Major comments: The rationale on which the study was based is not strong enough - Why is it necessary to understand the role of social relationships in risk-behaviors among adolescents?

Response: Thank you for your comment. By adding a sentence from your previous advise, I believe this problem also might have been solved. 

Revised manuscript, line 50~52: Furthermore, according to the pervious study, peer differentiation strongly influences psychosocial maturity of adolescents, which can also eventually lead to problematic behaviors.

Data and study population:

Major comments: The total KYRBS survey participants comprised 167,099 school students, but approximately half of the survey participants were excluded from the study sample. Please provide the point-by-point exclusion criteria – In what process, did the survey participants drop out the most and why?

Response: Thank you for your comment. I totally agree with your comment. I carefully examined where the exclusion has been made, and added a sentence to the manuscript explaining it. The added sentence is “ next, among 166,590, 87,532 remained due to the missing data from the variable ‘educational level of father and mother.’ Finally, there was no missing data in other covariates such as economic level, subjective health status, stress level, and BMI, leading the remaining 87,532 as the final study sample size.”

Revised manuscript, line 88~91: Next, among 166,590, 87,532 remained due to the missing data from the variable ‘educational level of father and mother.’ Finally, there was no missing data in other covariates such as economic level, subjective health status, stress level, and BMI, leading the remaining 87,532 as the final study sample size.

Variables:

Major comments: Please clarify the dependent variable of the study. How is the current smoking and drinking status measured? Was it examined? or based on questionnaire?

Response: Thank you for your comment. I have written in the limitation that the smoking and drinking status was based on the survey. To enhance an understanding of the study, I also have added a following sentence in method section

Revised manuscript, line 112~113: All answers were based on self-reported measures.

Major comments: The official outbreak of COVID-19 assigned by WHO is Feb 2020. Is the period of KYRBS data collection appropriate for the study?

Response: Thank you for your comment. Even though the official outbreak of COVID-19 pandemic assigned by WHO is Feb 2020, the first recognition of the pandemic was in December 2019, which means that the COVID-19 has impacted Koreans from the very beginning of 2020. In addition, I could find various other researches which considered 2020 as a start year of COVID-19. Still, I appreciate your careful advise. 

Table 1:

Major comments: In the statistical analysis section, it is stated that the chi-square test was conducted to analyze the general characteristics of the study population. However, Table 1 does not present the result of this test. Please provide the result of chi-square test for independence. Is the distribution of categorized group differ from one another with the significant P-value?

Response: Thank you for your comment. Sorry for making a confusion and thank you for pointing out our mistake. We added a section which tells the p-value of result of chi-square test in the right side of table 1. Again, I appreciate your advise. 

Table 2:

Major comments: Please specify the performed test in the title of Tables 2 & 3.

Response: Thank you for your comment. I totally agree with your comment. Following your advise, I changed the title of Tables 2,3, and 4 to “Table 2. Logistic Regression Results on Drinking for 2019 vs 2020, 2019 vs 2021 Participants,” “Table 3. Logistic Regression Results on Smoking for 2019 vs 2020, 2019 vs 2021 Participants,” and “Table 4. Logistic Regression Results on Substance Uses for 2019 vs 2020, 2019 vs 2021 Participants By Sex and Scholastic Performance,” accordingly. 

Major comments: The title of the study and the result do not align. I recommend estimating the OR for “Current Drinking Status” and “Current Smoking Status.” With the year 2019 served as reference, please provide OR for 2020 and 2021 – How much “less” likely the drinking or smoking is likely to occur relative to 2019? Please update the table and text accordingly.

Response: Thank you for your comment. I totally agree with your comment. Following your advise, we have changed the result to show how much less likely the drinking or smoking would occur. Also, other parts in the manuscript were changed accordingly. 

Discussion:

Major comments: The statement “To our knowledge, this is the first study to present the changes of alcohol and tobacco consumption during the COVID-19 pandemic of Korean teenagers” is not true. Please remove the statement or rephrase it to “there is little empirical evidence on …”

Response: Thank you for your comment. I totally agree with your comment. I changed the sentence into “There is little empirical evidence on the changes of alcohol and tobacco consumption during the COVID-19 pandemic of Korean teenagers.”

 Revised manuscript, line 174~175: There is little empirical evidence on the changes of alcohol and tobacco consumption during the COVID-19 pandemic of Korean teenagers.

Major comments: Please cite for more information. The discussion part can be better written with proper explanations and justifications of the data.

Response: Thank you for your comment. Since all the necessary information was cited in the discussion section, I have added an explanation and a citation on the introduction instead. Citation is very important and thank you very much for reminding me of it. 

Response to Editor’s comments

Minor Revision: Please consider using a standard terminology, such as, "alcohol use" than "drinking" throughout the manuscript.

Response: Thank you for your comment. We totally agree with your comment. We changed every single term used as “drinking” into more standard terms such as “alcohol use” and “taking alcohol.”

Minor comments: I also found that the manuscript needs to undergo a through language editing for better clarity of the sentences.

Response: Thank you for your comment. We totally agree with your comment. Also, thank you very much for kindly telling me the parts where the language editing is required. I changed all the necessary parts following your sincere advises. 

Minor comments: To assess the difference across the time period (2019 vs 2020 vs 2021), the p-value for the time trend should be calculated. The Cochran-Armitage and Chi-squared test for linear and non-linear time trends, respectively, can be calculated. 

Response: Thank you for your comment. I totally agree with your comment. I added a sentence, “Cochran-Armitage tests were used to test linear time-trend estimates, while Chi-squared tests were conducted to assess the association of demographic and socioeconomic characteristics of the study population with their substance use.” (revised manuscript, line 126~128) Also, the test was performed accordingly. Thank you for your meaningful comment.

Revised manuscript, line 126~128: Cochran-Armitage tests were used to test linear time-trend estimates, while Chi-squared tests were conducted to assess the association of demographic and socioeconomic characteristics of the study population with their substance use

Minor comments: In Table 4, please change the ORs as per the results discussed. Changing the reference category from "Yes" to "No" would suffice. That would provide lower ORs in 2020/2021 compared to 2019.

Response: Thank you for your comment. We totally agree with your comment. We have changed the results of the subgroup analyses accordingly. Also, the subsequent changes in the results section of manuscript was also applied.

Minor comments: Other minor editing required has been highlighted in the attached manuscripts use track change mode and adding comments.

Response: Thank you for your comment. Thank you for correcting my grammatical error. We have changed every point that you have told us. I appreciate your kind consideration.

---

## [Editor Report · Decision Letter 2]

7 Dec 2022

PONE-D-22-20225R2Comparison of Alcohol Consumption and Tobacco Use Among Korean Adolescents Before and During the COVID-19 PandemicPLOS ONE

Dear Dr. Jeong,

Thank you for submitting your manuscript to PLOS ONE. After careful consideration, we feel that it has merit but does not fully meet PLOS ONE’s publication criteria as it currently stands. Therefore, we invite you to submit a revised version of the manuscript that addresses the points raised during the review process.

We look forward to receiving your revised manuscript.

Kind regards,

Chandan Kumar, Ph.D.

Academic Editor

PLOS ONE

Additional Editor Comments:

1. Author has mentioned "Cochran-Armitage tests were used to test linear time-trend estimates, while Chi-squared tests were conducted to assess the association of demographic and socioeconomic characteristics of the study population with their substance use" in the revised manuscript. However, the results in the Table has not been modified. I don't see any change in the Table-1 compared to the previous one. Please report the modified result in Table 1.

2. If the purpose of this paper and the multivariate analysis is just to show the adjusted ORs between periods (2019 vs 2020) and (2019 vs 2021), there is no need to show the estimates for other variables in the table. Author can just mention in the note that the estimates are adjusted for these variables. The detailed Table can be kept in the Supplementary file.

3. The manuscript needs a thorough language (English) editing. I have given a few examples while going through the manuscript, but those are not not the exhaustive one. There is still adequate scope to standardize the language for better clarity to the international audience. Please go through the editing done in the manuscript (in Track Change mode) and the additional comments mentioned in the doc.

---

## [Author Response · Author response to Decision Letter 2]

12 Jan 2023

Revision Note for MS ID: PONE-D-22-20225

Comparison of Alcohol Consumption and Tobacco Use Among Korean Adolescents Before and During the COVID-19 Pandemic

(PONE-D-22-20225)’

I was pleased to have the opportunity to revise my paper. In revising the paper, I have carefully considered your comments and suggestions. As instructed, I have attempted to explain the changes made in reaction to all of the reviewers and academic editor’s comments. The comments were very helpful overall, and I appreciate the constructive feedback on my original submission. After addressing the issues raised, I feel the quality of the paper has greatly improved and I hope you agree. My response to each comment is as follows, and I attach a revision note with the highlighted, revised sections of the manuscript. Again, thank you for the valuable and helpful comments.

Response to Reviewer #1’s comments

The study is well thought-out and the manuscript is clear and concise, however I do have some minor comments that can be taken into account before the paper is further considered for publication.

Thank you for your great efforts in reviewing my manuscript. After the review, I totally revised the manuscript. 

Minor Revision: Please revise the results section in the abstract once again. There are terms such as ‘ex-smoker’, ‘single smoker’ and ‘dual smoker’ used in the abstract which cannot be found in the main manuscript.

Response: Thank you for your comment. We totally agree with your comment. We changed the conclusion part of abstract by changing the sentence from “This study found that ex-smoker and dual smoker had inverse associations with the COVID-19 pandemic among Korean adolescents.” into “This study found that alcohol consumption and tobacco use had inverse associations with the COVID-19 pandemic among Korean adolescents.” (revised manuscript, line 34~35) Thank you so much for your careful revision which allowed us to find our mistakes. 

Revised manuscript, line 34~35: This study found that alcohol consumption and tobacco use had inverse associations with the COVID-19 pandemic among Korean adolescents.

Minor comments: It would be good if there was more information regarding the smoking and drinking variables, such as the exact number of cigarettes smoked or drinks consumed. The author also mentioned in the discussion that adolescents smoked fewer cigarettes and drank less alcohol, which would be accurate if variables quantifying the amount of tobacco and alcohol used were included in the analysis, if available. Otherwise, I would suggest including it in the limitations of the study.

Response: Thank you for your comment. We totally agree with your comment. We did not consider exact frequency of smoking/drinking since the studying populations. We, however, defined current smokers/drinkers as those who drank/smoked at least once withing a past month which would be still very detrimental to adolescents. Also, following your kind advise, we added the following sentence in the limitations of the study. 

Revised manuscript, line 202~204: Lastly, adolescents were considered as current smokers and drinkers if they have smoked or drank at least once within the past thirty days. The exact number of cigarettes smoked and drinks consumed were not involved in the investigation. 

Minor comments: Table 4 shows the subgroup analysis results of risk behaviors and COVID-19 pandemic, stratified by sex and scholastic performance, yet there is no mention of the significance of these results in the discussion section. Please write a line or two regarding these points in the discussion.

Response: Thank you for your comment. I totally agree with your comment. I added a sentence, “According to the subgroup analysis results of risk behaviors and COVID-19 pandemic in table 4, such peer pressure must have affected adolescents equally regardless of sex and educational performances.” (revised manuscript, line 184~186) Thank you for your meaningful comment.

Revised manuscript, line 184~186: According to the subgroup analysis results of risk behaviors and COVID-19 pandemic in table 4, such peer pressure must have affected adolescents equally regardless of sex and educational performances.

Minor comments: Since the author mentioned stress as a possible factor behind heightened alcohol and tobacco use in the discussion, is it possible to include ‘stress level’ in the subgroup analysis as well and check whether it is statistically meaningful?

Response: Thank you for your comment. We totally agree with your comment. have conducted a subgroup analysis including ‘stress level,’ but the result did not necessarily assist my comment on the reasonings for alcohol and tobacco use; no matter the stress levels every adolescent showed lower risk of smoking/drinking. Still, we appreciate your meaningful comment and providing me a chance to more deeply investigate our study.

Minor comments: “Reduced peer pressure and importance of peer relationships from the school closure could have stopped youths from conducting unnecessarily risk behaviors.” Please change ‘unnecessarily’ to ‘unnecessary’.

Response: Thank you for your comment. Thank you for correcting my grammatical error. The sentence has been changed as the below. 

Revised manuscript, line 182~184: Reduced peer pressure and importance of peer relationships from the school closure could have stopped youths from conducting unnecessary risk behaviors.

Response to Reviewer #2’s comments

This is an important study that compares the current smoking/drinking behavior of adolescents before and after the outbreak of the pandemic. The data source used in the study is also interesting, and likely to provide insight into the health effect of COVID-19. Overall, the manuscript is interesting and suitable for the journal's aims and scopes.

Thank you for your great efforts in reviewing our manuscript. I appreciate your comments and fully understand your concerns for the problems you have mentioned.

Title:

Major comments: The title did not adequately represent the methods and findings. In abstract, the outcome measures are current smoking and current drinking status. Whereas, the outcome measures written in the result are presented as “Non-Drinker” or “Non-Smoker.” Please update the title accordingly.

Response: Thank you for your comment. I totally understand what you mean. I changed the outcome measures written in the result as “Current Smoker” and “Current Drinker” in order to match the current title. 

Introduction:

Major comments: Please clarify the background section further.

Response: Thank you for your comment. We totally agree with your comment. We clarified background section by adding a following sentence: “As of 31 March 2020, a total of 9786 confirmed cases with COVID-19 have been reported in South Korea as well..” smoking on dyslipidemia in South Korean adults.” (revised manuscript, line 40~42) 

Major comments: What is the definition of adolescence? To justify the need for this study, please discuss the plausibility – how is the stage of adolescence associated with risk behaviors? Why are they vulnerable to peer-pressure? Evidence regarding psychosocial development in adolescence is required.

Response: Thank you for your comment. I totally agree with your comment. I added a sentence which can justify how the stage of adolescence is associated with risk behaviors as follows: “Furthermore, according to the pervious study, peer differentiation strongly influences psychosocial maturity of adolescents, which can also eventually lead to problematic behaviors.” Furthermore, regarding the definition of adolescence, further description has been added in the method section. Thank you again for your meaningful advise. 

Revised manuscript, line 50~52: Furthermore, according to the pervious study, peer differentiation strongly influences psychosocial maturity of adolescents, which can also eventually lead to problematic behaviors.

Major comments: The rationale on which the study was based is not strong enough - Why is it necessary to understand the role of social relationships in risk-behaviors among adolescents?

Response: Thank you for your comment. By adding a sentence from your previous advise, I believe this problem also might have been solved. 

Revised manuscript, line 50~52: Furthermore, according to the pervious study, peer differentiation strongly influences psychosocial maturity of adolescents, which can also eventually lead to problematic behaviors.

Data and study population:

Major comments: The total KYRBS survey participants comprised 167,099 school students, but approximately half of the survey participants were excluded from the study sample. Please provide the point-by-point exclusion criteria – In what process, did the survey participants drop out the most and why?

Response: Thank you for your comment. I totally agree with your comment. I carefully examined where the exclusion has been made, and added a sentence to the manuscript explaining it. The added sentence is “ next, among 166,590, 87,532 remained due to the missing data from the variable ‘educational level of father and mother.’ Finally, there was no missing data in other covariates such as economic level, subjective health status, stress level, and BMI, leading the remaining 87,532 as the final study sample size.”

Revised manuscript, line 88~91: Next, among 166,590, 87,532 remained due to the missing data from the variable ‘educational level of father and mother.’ Finally, there was no missing data in other covariates such as economic level, subjective health status, stress level, and BMI, leading the remaining 87,532 as the final study sample size.

Variables:

Major comments: Please clarify the dependent variable of the study. How is the current smoking and drinking status measured? Was it examined? or based on questionnaire?

Response: Thank you for your comment. I have written in the limitation that the smoking and drinking status was based on the survey. To enhance an understanding of the study, I also have added a following sentence in method section

Revised manuscript, line 112~113: All answers were based on self-reported measures.

Major comments: The official outbreak of COVID-19 assigned by WHO is Feb 2020. Is the period of KYRBS data collection appropriate for the study?

Response: Thank you for your comment. Even though the official outbreak of COVID-19 pandemic assigned by WHO is Feb 2020, the first recognition of the pandemic was in December 2019, which means that the COVID-19 has impacted Koreans from the very beginning of 2020. In addition, I could find various other researches which considered 2020 as a start year of COVID-19. Still, I appreciate your careful advise. 

Table 1:

Major comments: In the statistical analysis section, it is stated that the chi-square test was conducted to analyze the general characteristics of the study population. However, Table 1 does not present the result of this test. Please provide the result of chi-square test for independence. Is the distribution of categorized group differ from one another with the significant P-value?

Response: Thank you for your comment. Sorry for making a confusion and thank you for pointing out our mistake. We added a section which tells the p-value of result of chi-square test in the right side of table 1. Again, I appreciate your advise. 

Table 2:

Major comments: Please specify the performed test in the title of Tables 2 & 3.

Response: Thank you for your comment. I totally agree with your comment. Following your advise, I changed the title of Tables 2,3, and 4 to “Table 2. Logistic Regression Results on Drinking for 2019 vs 2020, 2019 vs 2021 Participants,” “Table 3. Logistic Regression Results on Smoking for 2019 vs 2020, 2019 vs 2021 Participants,” and “Table 4. Logistic Regression Results on Substance Uses for 2019 vs 2020, 2019 vs 2021 Participants By Sex and Scholastic Performance,” accordingly. 

Major comments: The title of the study and the result do not align. I recommend estimating the OR for “Current Drinking Status” and “Current Smoking Status.” With the year 2019 served as reference, please provide OR for 2020 and 2021 – How much “less” likely the drinking or smoking is likely to occur relative to 2019? Please update the table and text accordingly.

Response: Thank you for your comment. I totally agree with your comment. Following your advise, we have changed the result to show how much less likely the drinking or smoking would occur. Also, other parts in the manuscript were changed accordingly. 

Discussion:

Major comments: The statement “To our knowledge, this is the first study to present the changes of alcohol and tobacco consumption during the COVID-19 pandemic of Korean teenagers” is not true. Please remove the statement or rephrase it to “there is little empirical evidence on …”

Response: Thank you for your comment. I totally agree with your comment. I changed the sentence into “There is little empirical evidence on the changes of alcohol and tobacco consumption during the COVID-19 pandemic of Korean teenagers.”

 Revised manuscript, line 174~175: There is little empirical evidence on the changes of alcohol and tobacco consumption during the COVID-19 pandemic of Korean teenagers.

Major comments: Please cite for more information. The discussion part can be better written with proper explanations and justifications of the data.

Response: Thank you for your comment. Since all the necessary information was cited in the discussion section, I have added an explanation and a citation on the introduction instead. Citation is very important and thank you very much for reminding me of it. 

Response to Editor’s comments

Minor Revision: Please consider using a standard terminology, such as, "alcohol use" than "drinking" throughout the manuscript.

Response: Thank you for your comment. We totally agree with your comment. We changed every single term used as “drinking” into more standard terms such as “alcohol use” and “taking alcohol.”

Minor comments: I also found that the manuscript needs to undergo a through language editing for better clarity of the sentences.

Response: Thank you for your comment. We totally agree with your comment. Also, thank you very much for kindly telling me the parts where the language editing is required. I changed all the necessary parts following your sincere advises. 

Minor comments: To assess the difference across the time period (2019 vs 2020 vs 2021), the p-value for the time trend should be calculated. The Cochran-Armitage and Chi-squared test for linear and non-linear time trends, respectively, can be calculated. 

Response: Thank you for your comment. I totally agree with your comment. I added a sentence, “Cochran-Armitage tests were used to test linear time-trend estimates, while Chi-squared tests were conducted to assess the association of demographic and socioeconomic characteristics of the study population with their substance use.” (revised manuscript, line 126~128) Also, the test was performed accordingly. Thank you for your meaningful comment.

Revised manuscript, line 126~128: Cochran-Armitage tests were used to test linear time-trend estimates, while Chi-squared tests were conducted to assess the association of demographic and socioeconomic characteristics of the study population with their substance use

Minor comments: In Table 4, please change the ORs as per the results discussed. Changing the reference category from "Yes" to "No" would suffice. That would provide lower ORs in 2020/2021 compared to 2019.

Response: Thank you for your comment. We totally agree with your comment. We have changed the results of the subgroup analyses accordingly. Also, the subsequent changes in the results section of manuscript was also applied.

Minor comments: Other minor editing required has been highlighted in the attached manuscripts use track change mode and adding comments.

Response: Thank you for your comment. Thank you for correcting my grammatical error. We have changed every point that you have told us. I appreciate your kind consideration. 

Major comments: Author has mentioned "Cochran-Armitage tests were used to test linear time-trend estimates, while Chi-squared tests were conducted to assess the association of demographic and socioeconomic characteristics of the study population with their substance use" in the revised manuscript. However, the results in the Table has not been modified. I don't see any change in the Table-1 compared to the previous one. Please report the modified result in Table 1.

Response: Thank you for your comment. It is true that the results were not modified, but it was not because have not changed the results, but because the two tests presented the same results. I have applied Cochran Armitage tests for the linear trends and specified those that were applied. I have attached a screen shot of the SAS results of main dependent variables CSMK( current smoking status) and DRINK (current alcohol use status) for your information. 

Major comments: If the purpose of this paper and the multivariate analysis is just to show the adjusted ORs between periods (2019 vs 2020) and (2019 vs 2021), there is no need to show the estimates for other variables in the table. Author can just mention in the note that the estimates are adjusted for these variables. The detailed Table can be kept in the Supplementary file.

Response: Thank you for your comment. We totally agree with your comment. Following your advises, we have erased the estimates for other variables in the table 2 and 3. The original tables were kept in the another file called “supplementary tables.” Also, I have mixed the results from the two tables into one, since only the adjusted ORs between the periods were presented.

Major comments: The manuscript needs a thorough language (English) editing. I have given a few examples while going through the manuscript, but those are not the exhaustive one. There is still adequate scope to standardize the language for better clarity to the international audience. Please go through the editing done in the manuscript (in Track Change mode) and the additional comments mentioned in the doc.

Response: Thank you for your comment. We totally agree with your comment. We have changed the examples which you gave us as well as few subtle errors through the manuscript. Thank you so much for your help in fixing the grammatical errors and sorry for providing you the unnecessary inconvenience.

---

## [Editor Report · Decision Letter 3]

24 Jan 2023

PONE-D-22-20225R3Comparison of Alcohol Consumption and Tobacco Use Among Korean Adolescents Before and During the COVID-19 PandemicPLOS ONE

Dear Dr. Jeong,

Thank you for submitting your manuscript to PLOS ONE. After careful consideration, we feel that it has merit but does not fully meet PLOS ONE’s publication criteria as it currently stands. Therefore, we invite you to submit a revised version of the manuscript that addresses the points raised during the review process.

Academic Editor's Comments:

1. The time-trend analysis has not been done. I don't see any change in the table since the first submission of the manuscript. I'd asked for providing p-values for time trends. If you are not getting it, please follow the following tutorial:

https://www.lexjansen.com/pharmasug/2007/sp/SP05.pdf. You'll have to first convert all the factor variables into dummies and for each of the categories you will compare the proportion of your outcome variables (e.g., alcohol use and tobacco consumption) over time. Since Table 1 presents only the sample characteristics, You need to prepare a separate Table presenting bi-variate statistics. In Table-1, there is no need to provide p-values; what these p-values are presenting is not understood.

2. Also see the minor editing done in the attached manuscript.

We look forward to receiving your revised manuscript.

Kind regards,

Chandan Kumar, Ph.D.

Academic Editor

PLOS ONE

---

## [Author Response · Author response to Decision Letter 3]

7 Mar 2023

I was pleased to have the opportunity to revise my paper. In revising the paper, I have carefully considered your comments and suggestions. As instructed, I have attempted to explain the changes made in reaction to all of the reviewers and academic editor’s comments. The comments were very helpful overall, and I appreciate the constructive feedback on my original submission. After addressing the issues raised, I feel the quality of the paper has greatly improved and I hope you agree. My response to each comment is as follows, and I attach a revision note with the highlighted, revised sections of the manuscript. Again, thank you for the valuable and helpful comments.

Response to Reviewer #1’s comments

The study is well thought-out and the manuscript is clear and concise, however I do have some minor comments that can be taken into account before the paper is further considered for publication.

Thank you for your great efforts in reviewing my manuscript. After the review, I totally revised the manuscript. 

Minor Revision: Please revise the results section in the abstract once again. There are terms such as ‘ex-smoker’, ‘single smoker’ and ‘dual smoker’ used in the abstract which cannot be found in the main manuscript.

Response: Thank you for your comment. We totally agree with your comment. We changed the conclusion part of abstract by changing the sentence from “This study found that ex-smoker and dual smoker had inverse associations with the COVID-19 pandemic among Korean adolescents.” into “This study found that alcohol consumption and tobacco use had inverse associations with the COVID-19 pandemic among Korean adolescents.” (revised manuscript, line 34~35) Thank you so much for your careful revision which allowed us to find our mistakes. 

Revised manuscript, line 34~35: This study found that alcohol consumption and tobacco use had inverse associations with the COVID-19 pandemic among Korean adolescents.

Minor comments: It would be good if there was more information regarding the smoking and drinking variables, such as the exact number of cigarettes smoked or drinks consumed. The author also mentioned in the discussion that adolescents smoked fewer cigarettes and drank less alcohol, which would be accurate if variables quantifying the amount of tobacco and alcohol used were included in the analysis, if available. Otherwise, I would suggest including it in the limitations of the study.

Response: Thank you for your comment. We totally agree with your comment. We did not consider exact frequency of smoking/drinking since the studying populations. We, however, defined current smokers/drinkers as those who drank/smoked at least once withing a past month which would be still very detrimental to adolescents. Also, following your kind advise, we added the following sentence in the limitations of the study. 

Revised manuscript, line 202~204: Lastly, adolescents were considered as current smokers and drinkers if they have smoked or drank at least once within the past thirty days. The exact number of cigarettes smoked and drinks consumed were not involved in the investigation. 

Minor comments: Table 4 shows the subgroup analysis results of risk behaviors and COVID-19 pandemic, stratified by sex and scholastic performance, yet there is no mention of the significance of these results in the discussion section. Please write a line or two regarding these points in the discussion.

Response: Thank you for your comment. I totally agree with your comment. I added a sentence, “According to the subgroup analysis results of risk behaviors and COVID-19 pandemic in table 4, such peer pressure must have affected adolescents equally regardless of sex and educational performances.” (revised manuscript, line 184~186) Thank you for your meaningful comment.

Revised manuscript, line 184~186: According to the subgroup analysis results of risk behaviors and COVID-19 pandemic in table 4, such peer pressure must have affected adolescents equally regardless of sex and educational performances.

Minor comments: Since the author mentioned stress as a possible factor behind heightened alcohol and tobacco use in the discussion, is it possible to include ‘stress level’ in the subgroup analysis as well and check whether it is statistically meaningful?

Response: Thank you for your comment. We totally agree with your comment. have conducted a subgroup analysis including ‘stress level,’ but the result did not necessarily assist my comment on the reasonings for alcohol and tobacco use; no matter the stress levels every adolescent showed lower risk of smoking/drinking. Still, we appreciate your meaningful comment and providing me a chance to more deeply investigate our study.

Minor comments: “Reduced peer pressure and importance of peer relationships from the school closure could have stopped youths from conducting unnecessarily risk behaviors.” Please change ‘unnecessarily’ to ‘unnecessary’.

Response: Thank you for your comment. Thank you for correcting my grammatical error. The sentence has been changed as the below. 

Revised manuscript, line 182~184: Reduced peer pressure and importance of peer relationships from the school closure could have stopped youths from conducting unnecessary risk behaviors.

Response to Reviewer #2’s comments

This is an important study that compares the current smoking/drinking behavior of adolescents before and after the outbreak of the pandemic. The data source used in the study is also interesting, and likely to provide insight into the health effect of COVID-19. Overall, the manuscript is interesting and suitable for the journal's aims and scopes.

Thank you for your great efforts in reviewing our manuscript. I appreciate your comments and fully understand your concerns for the problems you have mentioned.

Title:

Major comments: The title did not adequately represent the methods and findings. In abstract, the outcome measures are current smoking and current drinking status. Whereas, the outcome measures written in the result are presented as “Non-Drinker” or “Non-Smoker.” Please update the title accordingly.

Response: Thank you for your comment. I totally understand what you mean. I changed the outcome measures written in the result as “Current Smoker” and “Current Drinker” in order to match the current title. 

Introduction:

Major comments: Please clarify the background section further.

Response: Thank you for your comment. We totally agree with your comment. We clarified background section by adding a following sentence: “As of 31 March 2020, a total of 9786 confirmed cases with COVID-19 have been reported in South Korea as well..” smoking on dyslipidemia in South Korean adults.” (revised manuscript, line 40~42) 

Major comments: What is the definition of adolescence? To justify the need for this study, please discuss the plausibility – how is the stage of adolescence associated with risk behaviors? Why are they vulnerable to peer-pressure? Evidence regarding psychosocial development in adolescence is required.

Response: Thank you for your comment. I totally agree with your comment. I added a sentence which can justify how the stage of adolescence is associated with risk behaviors as follows: “Furthermore, according to the pervious study, peer differentiation strongly influences psychosocial maturity of adolescents, which can also eventually lead to problematic behaviors.” Furthermore, regarding the definition of adolescence, further description has been added in the method section. Thank you again for your meaningful advise. 

Revised manuscript, line 50~52: Furthermore, according to the pervious study, peer differentiation strongly influences psychosocial maturity of adolescents, which can also eventually lead to problematic behaviors.

Major comments: The rationale on which the study was based is not strong enough - Why is it necessary to understand the role of social relationships in risk-behaviors among adolescents?

Response: Thank you for your comment. By adding a sentence from your previous advise, I believe this problem also might have been solved. 

Revised manuscript, line 50~52: Furthermore, according to the pervious study, peer differentiation strongly influences psychosocial maturity of adolescents, which can also eventually lead to problematic behaviors.

Data and study population:

Major comments: The total KYRBS survey participants comprised 167,099 school students, but approximately half of the survey participants were excluded from the study sample. Please provide the point-by-point exclusion criteria – In what process, did the survey participants drop out the most and why?

Response: Thank you for your comment. I totally agree with your comment. I carefully examined where the exclusion has been made, and added a sentence to the manuscript explaining it. The added sentence is “ next, among 166,590, 87,532 remained due to the missing data from the variable ‘educational level of father and mother.’ Finally, there was no missing data in other covariates such as economic level, subjective health status, stress level, and BMI, leading the remaining 87,532 as the final study sample size.”

Revised manuscript, line 88~91: Next, among 166,590, 87,532 remained due to the missing data from the variable ‘educational level of father and mother.’ Finally, there was no missing data in other covariates such as economic level, subjective health status, stress level, and BMI, leading the remaining 87,532 as the final study sample size.

Variables:

Major comments: Please clarify the dependent variable of the study. How is the current smoking and drinking status measured? Was it examined? or based on questionnaire?

Response: Thank you for your comment. I have written in the limitation that the smoking and drinking status was based on the survey. To enhance an understanding of the study, I also have added a following sentence in method section

Revised manuscript, line 112~113: All answers were based on self-reported measures.

Major comments: The official outbreak of COVID-19 assigned by WHO is Feb 2020. Is the period of KYRBS data collection appropriate for the study?

Response: Thank you for your comment. Even though the official outbreak of COVID-19 pandemic assigned by WHO is Feb 2020, the first recognition of the pandemic was in December 2019, which means that the COVID-19 has impacted Koreans from the very beginning of 2020. In addition, I could find various other researches which considered 2020 as a start year of COVID-19. Still, I appreciate your careful advise. 

Table 1:

Major comments: In the statistical analysis section, it is stated that the chi-square test was conducted to analyze the general characteristics of the study population. However, Table 1 does not present the result of this test. Please provide the result of chi-square test for independence. Is the distribution of categorized group differ from one another with the significant P-value?

Response: Thank you for your comment. Sorry for making a confusion and thank you for pointing out our mistake. We added a section which tells the p-value of result of chi-square test in the right side of table 1. Again, I appreciate your advise. 

Table 2:

Major comments: Please specify the performed test in the title of Tables 2 & 3.

Response: Thank you for your comment. I totally agree with your comment. Following your advise, I changed the title of Tables 2,3, and 4 to “Table 2. Logistic Regression Results on Drinking for 2019 vs 2020, 2019 vs 2021 Participants,” “Table 3. Logistic Regression Results on Smoking for 2019 vs 2020, 2019 vs 2021 Participants,” and “Table 4. Logistic Regression Results on Substance Uses for 2019 vs 2020, 2019 vs 2021 Participants By Sex and Scholastic Performance,” accordingly. 

Major comments: The title of the study and the result do not align. I recommend estimating the OR for “Current Drinking Status” and “Current Smoking Status.” With the year 2019 served as reference, please provide OR for 2020 and 2021 – How much “less” likely the drinking or smoking is likely to occur relative to 2019? Please update the table and text accordingly.

Response: Thank you for your comment. I totally agree with your comment. Following your advise, we have changed the result to show how much less likely the drinking or smoking would occur. Also, other parts in the manuscript were changed accordingly. 

Discussion:

Major comments: The statement “To our knowledge, this is the first study to present the changes of alcohol and tobacco consumption during the COVID-19 pandemic of Korean teenagers” is not true. Please remove the statement or rephrase it to “there is little empirical evidence on …”

Response: Thank you for your comment. I totally agree with your comment. I changed the sentence into “There is little empirical evidence on the changes of alcohol and tobacco consumption during the COVID-19 pandemic of Korean teenagers.”

 Revised manuscript, line 174~175: There is little empirical evidence on the changes of alcohol and tobacco consumption during the COVID-19 pandemic of Korean teenagers.

Major comments: Please cite for more information. The discussion part can be better written with proper explanations and justifications of the data.

Response: Thank you for your comment. Since all the necessary information was cited in the discussion section, I have added an explanation and a citation on the introduction instead. Citation is very important and thank you very much for reminding me of it. 

Response to Editor’s comments

Major Revision: The time-trend analysis has not been done. I don't see any change in the table since the first submission of the manuscript. I'd asked for providing p-values for time trends. You'll have to first convert all the factor variables into dummies and for each of the categories you will compare the proportion of your outcome variables (e.g., alcohol use and tobacco consumption) over time. Since Table 1 presents only the sample characteristics, You need to prepare a separate Table presenting bi-variate statistics. In Table-1, there is no need to provide p-values; what these p-values are presenting is not understood.

Response: Thank you for your comment. Sorry for providing inconvenience. Following your instruction, we have made dummy variables for each categories and made separate table 2 and 3 for tobacco consumption and alcohol use. Also, a paragraph has been added respectively. Thank you so much for you help in further developing our manuscript. 

Minor comments: Also see the minor editing done in the attached manuscript.

Response: Thank you for your comment. We totally agree with your comment. Following your advises, we have applied the minor editing. Thank you so much for your kind help.

---

## [Editor Report · Decision Letter 4]

9 Mar 2023

Comparison of Alcohol Consumption and Tobacco Use Among Korean Adolescents Before and During the COVID-19 Pandemic

PONE-D-22-20225R4

Dear Dr. Jeong,

We’re pleased to inform you that your manuscript has been judged scientifically suitable for publication and will be formally accepted for publication once it meets all outstanding technical requirements.

Kind regards,

Chandan Kumar, Ph.D.

Academic Editor

PLOS ONE

---

## [Editor Report · Acceptance letter]

15 Mar 2023

PONE-D-22-20225R4 

Comparison of Alcohol Consumption and Tobacco Use among Korean Adolescents before and during the COVID-19 Pandemic 

Dear Dr. Jeong:

I'm pleased to inform you that your manuscript has been deemed suitable for publication in PLOS ONE. Congratulations! Your manuscript is now with our production department. 

Kind regards, 

on behalf of

Dr. Chandan Kumar 

Academic Editor

PLOS ONE